# Insights into Molecular Mechanism of Secondary Xylem Rapid Growth in *Salix psammophila*

**DOI:** 10.3390/plants14030459

**Published:** 2025-02-05

**Authors:** Hongxia Qiao, Yunhan Wang, Lin Shi, Ruiping Wang, Yeru Yang, Dongshan Wei, Yingjie Li, Kairui Chao, Li Jia, Guiming Liu, Fengqiang Yu, Jiewei Zhang, Haifeng Yang

**Affiliations:** 1Forestry College, Inner Mongolia Agricultural University, Hohhot 010018, China; qhx1934362822@126.com (H.Q.); wdsh0812@imau.edu.cn (D.W.); liyingjie2000@126.com (Y.L.); 17190087001@imau.edu.cn (K.C.); 2Beijing Academy of Agriculture and Forestry Sciences, Beijing 100097, China; 202330123123@bua.edu.cn (Y.W.); mingguiliu@aliyun.com (G.L.); 3Beijing Key Laboratory of Agricultural Genetic Resources and Biotechnology, Institute of Biotechnology, Beijing 100097, China; 4College of Bioscience and Resources Environment, Beijing University of Agriculture, Bejing 102206, China; 5Ordos Research Institute of Forestry and Grassland Science, Ordos 017000, China; shilin_323@163.com; 6Inner Mongolia Ordos Forestry and Grassland Business Development Center, Ordos 017000, China; erdoszmz@126.com (R.W.); 13947734335@126.com (Y.Y.); jiali4123@126.com (L.J.)

**Keywords:** *Salix psammophila*, secondary growth, secondary xylem, lignin, caffeoyl shikimate esterase

## Abstract

Salix psammophila C. Wang & C. Y. Yang is an important windbreak and sand-fixing shrub species in Northwest China, with excellent characteristics such as resistance to drought, wind, and sand. *S. psammophila* needs to be stubbed flat after several years of growth to continue to grow, otherwise, its growth rate will slow down and even begin to die. To understand the genetic regulatory mechanism of secondary growth in *S. psammophila*, cell structure and transcriptome analysis were performed on the secondary xylem and secondary phloem of stems. The results showed that the secondary xylem and the secondary phloem of *S. psammophila* were well developed at 1, 2, and 3-year-old stages, and the secondary growth changes mainly occurred in the secondary xylem at the 2 to 3-year-old stage, with a faster growth rate. The *CSE2* and *CSE1* genes that regulate CSE (caffeoyl shikimate esterase) have high sequence similarity (92% and 93%) with the *CSE2* and *CSE1* genes of the genus *Populus*, respectively, and regulate lignin biosynthesis. Notably, the expression levels of these two genes decreased in the secondary xylem of 3-year-old *S. psammophila*, indicating that the rapid growth of *S. psammophila* may be related to lignin biosynthesis. Weighted gene co-expression network analysis (WGCNA) was utilized to screen candidate TFs and genes involved in the secondary growth processes of *S. psammophila*, which were categorized into six co-expression modules. A total of 79 genes were selected from these co-expression modules, and co-expression network maps of the genes were constructed. The results indicate that the secondary growth of *S. psammophila* was regulated by a TF regulatory network. Interestingly, PLATZ TFs were involved in the rapid secondary growth and stress tolerance in *S. psammophila*. This hints that *S. psammophila* may promote secondary growth by increasing stress tolerance.

## 1. Introduction

*Salix psammophila* C. Wang & C. Y. Yang is a species with excellent characteristics such as resistance to drought, extreme temperatures, wind, sand, etc. [1]. It serves as a crucial windbreak and sand-fixing shrub in Northwest China. Additionally, it plays an important role in vegetation restoration and afforestation, thriving in Northern China’s arid and semi-arid desert regions [1,2,3]. The growth cycle of *S. psammophila* is relatively short, and to continue to grow vigorously, it needs to be pruned. Otherwise, the growth will gradually decrease, and even begin to die [4,5]. Wood is an economical and sustainable feedstock for bioenergy production [6]. After the flat stubble treatment, a significant number of branches are produced, which can be used as growth materials for the manufacturing of composite boards, paper, and activated charcoal [4,5,7]. The asexual *S. psammophila* ‘17-38’ is an excellent asexual fast-growing variety with an upright morphology and numerous short branchlets. In recent years, there has been increased research on *S. psammophila* due to its economic and ecological benefits [2,8,9,10,11]. However, the mechanism that regulates the secondary growth of *S. psammophila* remains unclear. This paper analyzed the cell structure and transcriptome of the secondary xylem and phloem from the stems of the asexual *S. psammophila* ‘17-38’ at 1, 2, and 3-year-old stages.

Secondary growth originates from the vascular cambium and is the process of thickening of woody stems. The vascular cambium generates a secondary xylem toward the interior and a secondary phloem toward the exterior [12,13]. Secondary growth (wood formation) is a programmed process that includes the formation of the vascular cambium and secondary cell wall (SCW), deposition of lignin, programmed cell death, and the creation of heartwood [14,15]. The xylem consists of xylem fibers and vessels, and after completing the initial expansion of the xylem fiber cells in the radial and longitudinal directions, it enters the synthesis of SCW, completes fiber differentiation, and finally enters programmed cell death and autolysis of cellular contents [16]. Cell wall thickening and lignification also occur in the phloem fiber cells of the phloem tissue formed by the differentiation of derivatives on the outer surface of the vascular cambium [17]. Lignin, cellulose, and hemicellulose are the main components of SCW [18,19]. Cellulose and hemicellulose are two polysaccharides that interact through hydrogen bonding to form the skeleton of the SCW [20,21]. Lignin is a complex polyphenolic compound that oxidizes p-coumaryl alcohol, coniferyl alcohol, and sinapyl alcohol in the SCW to create lignin free radicals. These radicals then combine and couple to form H-, G-, and S-lignin, which become incorporated into the skeleton of the SCW made up of polysaccharides and ultimately complete the biosynthesis of the SCW [18,19].

The process of secondary growth is the engine of wood formation, including the biosynthesis of cell walls, cellulose, and lignin. This process is regulated by various factors such as phytohormones, signaling peptides, and transcription factors (TFs) [6,22,23,24,25]. TFs of the MYB family are the largest class of TFs and are essential in plant SCW biosynthesis [26,27]. For example, the R2R3 TF *EgMYB1* is primarily expressed in differentiated xylem and binds specifically to the MBSIIG site in the promoters of genes involved in lignin biosynthesis. *EgMYB1* represses the expression of two essential genes involved in lignin production in *Eucalyptus* and serves as a negative regulator during the formation of the secondary wall [28,29]. The overexpression of the R2R3 MYB TF MYB189 in poplar and *Arabidopsis* inhibited SCW biosynthesis [30]. NAC domain TFs are specific to plants, regulate plant growth, development and stress responses, and are major switches in the SCW biosynthesis network [31,32]. Xylem NAC domain 1 (XND1) overexpression negatively regulates SCW deposition [33]. Johnsson et al. found that treatment of *Arabidopsis thaliana* and *Populus* stems with NAA and GA resulted in changes in the transcription of NAC domain TFs associated with SCW formation, affecting the rate of SCW deposition and wood formation [34].

Functional genomic information can be obtained more effectively and economically with next-generation sequencing technologies [35,36]. To determine the essential genes for *S. psammophila* secondary growth, we collected the secondary xylem and the secondary phloem of stems of *S. psammophila* grown at 1, 2, and 3-year-old. We combined transcriptome sequencing to analyze the expression of genes. The process of secondary growth in plants is regulated by a transcriptional regulatory network formed by transcription factor genes. The changes in this process occur mainly in the secondary xylem with large amounts of lignin deposition. The expression levels of transcription factor genes and genes regulating lignin synthesis during the secondary growth process of *S. psammophila* were explored by transcriptome analysis to elucidate the mechanism of rapid secondary growth in *S. psammophila*. This offers a more efficient method of investigating the genes involved in *S. psammophila* wood formation and advances our knowledge of the species’ secondary growth as well as the process of wood formation and development. This establishes a crucial theoretical basis for its use in the genetic enhancement of timber qualities and the development of superior breeding systems.

## 2. Results

### 2.1. Development Characteristics of Secondary Xylem and Phloem in Stem of S. psammophila

To investigate the changes in secondary growth of *S. psammophila*, the secondary xylem and phloem of 1, 2, and 3-year-old stems were collected for histological chemistry. The transverse section of the stem of *S. psammophila* consisted of the peridermis, cortex, phloem, cambium, xylem, and pith (Figure 1a). Peridermis is located on the outermost side of the stem. In the early years of growth, some layers of the primary cortex are still visible just below the peridermis. Adjacent to the primary cortex is the phloem, followed by the cambium zone, which is adjacent to the xylem. The xylem has a large number of randomly distributed vessels. And the pith is located in the center of the *S. psammophila* stem. The secondary xylem and phloem are well differentiated at different stages of growth and development. The secondary fiber walls of both the xylem and phloem and conducting elements of the xylem undergo significant lignification. The xylem and phloem have significant differences, and the degree of lignification of the secondary wall and phloem fibers gradually increases from 1 to 3-year-old.

### 2.2. RNA-seq Sequencing and Alignment to Reference Genome

A total of 18 transcriptome libraries were constructed using RNA-seq to analyze differentially expressed genes (DEGs) in different growth stages of the secondary xylem and secondary phloem of *S. psammophila*. Quality control of the raw data using Fastp (v0.24.3) produced more than 19 million clean reads of each sample, and all of them were above 98.43% for Q20, and 97.01% for Q30 (Appendix A). The squared correlation coefficient (r^2^) between biological replicate groups was >0.8, indicating that our RNA sequencing dataset is reliable for differential gene analysis.

Approximately 88.37% of the clean reads were mapped to the *S. purpurea* reference genome (Appendix A). To enhance the reliability of differential gene analysis, we performed inter-sample gene expression level correlation analysis and principal component analysis. The closer the correlation coefficient is to 1, the greater the similarity of expression patterns between samples. For specific program operations, we require that the r^2^ value between biological replicate samples be greater than 0.8. The correlation coefficients within and between group samples were calculated from the FPKM values of all genes in each sample and plotted as heat maps (Figure 1b). Principal component analysis (PCA) showed greater variability among 1, 2, and 3-year-old xylem samples and less variability among 1, 2, and 3-year-old phloem samples, with PC1 explaining a difference of 48.1% (Appendix A). Systematic clustering analysis supported these results, with 1-year-old xylem and 2-year-old xylem clustering into one group, 3-year-old xylem clustering into one group, and 1, 2, and 3-year-old phloem clustering into one group (Appendix A). Significant changes in the xylem occurred from 2 to 3-year-old during the secondary growth of *S. psammophila*.

### 2.3. Analysis of DEGs in Secondary Xylem and Phloem

To investigate the differential changes in the secondary xylem and the secondary phloem of *S. psammophila* at the stages of 1 to 2-year-old and 2 to 3-year-old, DEGs were analyzed in the secondary xylem and phloem of *S. psammophila* at the stages of 1, 2, and 3-year-old. Among these two growth stages of *S. psammophila*, the stage with the highest number of DEGs in the secondary xylem was the 2 to 3-year-old stage, with 4740 up-regulated DEGs and 3907 down-regulated DEGs (Figure 2a,b). The secondary phloem showed a decrease in the number of down-regulated DEGs and an increase in the number of DEGs in the secondary xylem (Figure 2c). This result is consistent with the results of the clustering analysis results, indicating that the changes in secondary growth and development of *S. psammophila* mainly occur in the secondary xylem of the 2 to 3-year-old stage.

### 2.4. GO Annotation of DEGs During Stem Xylem and Phloem Development

To further understand the biological functions of the DEGs, GO functional enrichment analyses were conducted on the DEGs in the secondary xylem and phloem of 1, 2, and 3-year-old *S. psammophila*. Among the detected DEGs, they were categorized into three main categories: biological process (BP), molecular function (MF), and cellular component (CC). A total of 52 GO entries were enriched in DEGs in the 1 and 2-year-old secondary xylem of *S. psammophila*, of which 16 were for biological processes, 7 for cellular components, and 29 for molecular functions (Appendix A). In the biological processes, up-regulated genes are mainly enriched in the sucrose metabolic process, the disaccharide metabolic process, and the oligosaccharide metabolic process. And down-regulated genes are mainly enriched in the protein phosphorylation, the phosphorylation, the ATP metabolic process, and the proteoly (Figure 3a). A total of 645 GO entries were enriched at this stage of 2 and 3-year-old, including 344 entries related to biological processes, 96 entries related to cellular components, and 205 entries related to molecular functions (Appendix A). In biological processes, up-regulated genes are mainly enriched in the regulation of the cellular process, the regulation of the macromolecule metabolic process, the regulation of the primary metabolic process, the regulation of the nitrogen compound metabolic process, the regulation of gene expression, the regulation of the cellular biosynthetic process, the regulation of the RNA metabolic process, the regulation of DNA-templated transcription, and the regulation of the nucleobase-containing compound metabolic process, while down-regulated genes are mainly enriched in vesicle-mediated transport, the establishment of localization, the glycoprotein metabolic process, protein glycosylation, cellular localization, the glycoprotein biosynthesis process, and protein localization (Figure 3b). Among the DEGs in the secondary phloem, a total of 286 GO entries, 175 biological process entries, 24 cellular component entries, and 87 molecular function entries were enriched in the 1 and 2-year-old secondary phloem DEGs (Appendix A). Down-regulated DEGs were mainly enriched in microtubule-based movement, mitotic cell cycle, nuclear division, and nuclear chromosome segregation (Figure 3c). A total of 125 GO entries, 62 biological process entries, 8 cellular component entries, and 55 molecular function entries were enriched at this stage of 2 and 3-year-old (Appendix A). Up-regulated DEGs were mainly enriched in mitochondrial respiratory chain complex III assembly, the inositol biosynthesis process, the phosphorus-containing compound metabolic process, the alcohol biosynthesis process, and the protein modification process (Figure 3d). These DEGs are mainly involved in cellular metabolism and protein synthesis and transport, and are related to programmed cell death and lignin synthesis during plant wood formation.

### 2.5. Analysis of Genes Related to Lignin Synthesis Pathways in Secondary Xylem and Secondary Phloem of Stems

The process of secondary growth is also the formation of wood, which is mainly composed of lignin, cellulose, and hemicellulose [6,12,37]. In this study, genes involved in lignin biosynthesis were analyzed. As shown in Figure 4, lignin synthesis is regulated by a series of enzymes, such as phenylalanine ammonia-lyase (PAL), cinnamate 4-hydroxylase (C4H), 4-hydroxycinnamoyl CoA: shikimate/quinate hydroxycinnamoyltransferase (HCT), 4-hydroxycinnamate: CoA ligase (4CL), cinnamoyl CoA reductase (CCR), caffeoyl shikimate esterase (CSE), caffeoyl CoA 3-O-methyltransferase (CCoAoMT), caffeate/5-hydroxyferulate 3-O-methyltransferase (COMT), and cinnamyl alcohol dehydrogenase (CAD). Five PAL genes, four C4H genes, and two CCoAoMT genes were up-regulated in the secondary xylem and down-regulated in the secondary phloem. In addition, the genes regulating CSE (*Sapur.003G032000*, *Sapur.001G143600*, *Sapur.010G177600*, *Sapur.008G029900*) showed the highest expression levels in the 1 and 2-year-old secondary xylem, and decreased or no expression in the 3-year-old secondary xylem. This analysis indicated that the lignin content of *S. psammophila* changed in the secondary xylem at 3-year-old.

### 2.6. Weighted Gene Co-Expression Network Analysis (WGCNA) Identified TFs Involved in Different Developmental Stages During Secondary Growth Stems

The 1147 genes in the secondary xylem and the secondary phloem of 1, 2, and 3-year-old *S. psammophila* stems were used in WGCNA to identify candidate TFs and genes involved in different stages of the secondary growth process. Scale free topology fitting showed that 15 is the optimum soft-threshold as the fitting index r^2^ > 0.85, while the mean connectivity is close to 0 (Appendix A). Thus, 15 is the soft-threshold for constructing co-expression networks. These genes were classified into six co-expression modules (Figure 5a). As shown in Figure 5b, the turquoise module was highly negatively correlated with the development of 1-year-old secondary xylem. The turquoise module includes MYB, NAC, HD-ZIP, Beta-Casp, and CASP gene families (Appendix A). The yellow module has 19 TFs, including TFs from the MYB, LBD, and PLATZ gene families, which were positively correlated with the development of the 1-year-old secondary phloem and have the strongest correlation, with a correlation coefficient (CC) of 0.73 (Appendix A).

To further understand the regulatory mechanisms of secondary growth in *S. psammophila*, we selected 79 genes from the MYB, NAC, and PLATZ gene families based on InterPro annotation results, and constructed co-expression network maps of the screened genes. PLATZ TFs, bHLH-MYC, and R2R3-MYB TFs, and NAC domain form a regulatory network (Figure 6a). Among them, *Sapur.006G098200*, which is related to PLATZ TFs, had the highest expression level in the secondary xylem of 2-year-old, possibly regulating the growth rate of plants by regulating the expression of genes (Figure 6b).

## 3. Discussion

The stem cell population of the vascular cambium belongs to the lateral meristematic tissue that produces secondary phloem (inner bark) and secondary xylem (wood) [12,38]. The xylem and phloem contain large amounts of lignin, cellulose, and hemicellulose. They are deposited in the secondary cell walls, leading to the thickening of the cell walls and wood formation [39,40]. Several genes were identified that are responsible for the formation of the cell wall, as well as for the synthesis of cellulose and hemicellulose [14]. For example, the overexpression of the *PagMYB128* gene in transgenic poplar increased the cellulose, hemicellulose, and lignin content of the wood [41]. In *Ricinus communis* L., the overexpression of *RcPAL* significantly enhanced PAL activity and lignin content, and identified it as a key gene in the lignin biosynthesis of *Ricinus communis* L. [42]. *EgMYB2* binds specifically to MYB consensus binding sites (cinnamoyl-coenzyme A reductase, CCR; and cinnamyl alcohol dehydrogenase, CAD) in the *cis*-regulatory regions of the promoters of lignin synthesis genes and further regulates SCW synthesis [43]. The significant differences in the xylem cells and phloem cells of the stem of *S. psammophila* indicate that there were significant changes in the secondary growth rate of *S. psammophila* during the 1, 2, and 3-year-old stages. We used RNA-seq to identify genes related to secondary growth and lignin synthesis and to analyze the regulatory mechanisms of secondary growth. The analysis of DEGs revealed that the growth changes in secondary growth of *S. psammophila* mainly occurred in the secondary xylem at the 2 to 3-year-old stage.

Lignin is a complex phenylpropanoid-derived polymer that is a major component of the SCW in plants [44]. The biosynthesis of lignin monomers begins with the deamination of phenylalanine. Following this, a series of hydroxylation, methylation, and reduction reactions occur, resulting in the synthesis of three monohydric alcohols: coumarinol, carvacrol, and pinacol. These alcohols are then polymerized to form the *p*-hydroxyphenyl (H), guaiacyl (G), and syringyl (S) units of lignin, respectively [45,46,47]. The lignin biosynthesis pathway requires the involvement of enzymes, and CSE regulates caffeic acid synthesis in the lignin biosynthesis pathway. However, not all plants contain CSE, and orthologs of CSE are lacking in some grass species [46]. Currently, orthologs of CSE are found in some fast-growing plants, such as poplar and *Eucalyptus* [48]. This research has identified genes involved in the regulation of CSE within the lignin synthesis pathway in the secondary xylem and secondary phloem of *S. psammophila* (*Sapur.003G032000*, *Sapur.001G143600*, *Sapur.010G177600*, and *Sapur.008G029900*). *Sapur.003G032000* and *Sapur.001G143600* have high similarity (92% and 93%) to the *CSE2* and *CSE1* gene sequences of the genus *Populus*, respectively (Appendix A). It was revealed that the functions of *CSE2* and *CSE1* genes, which play a role in poplar monolignol biosynthesis, are related to lignin biosynthesis [49]. The *Sapur.003G032000* and *Sapur.001G143600* genes are highly expressed in the secondary xylem at 1 and 2-year-old, but their expression levels decrease at 3-year-old. This phenomenon may be related to the secondary growth rate of *S. psammophila*, with a faster growth rate during the 2 to 3-year-old stage (Table 1). Fast-growing tree species provide a large amount of material for diversified biomass fuels [50]. As an important ecological tree species in Northwest China, the study of the rapid growth mechanism of *S. psammophila* is of great significance to meet the bioenergy demand in the region.

WGCNA can be used to find gene modules that are highly correlated with phenotypic traits and are widely used in RNA-seq analysis [14,51,52]. In this study, candidate TFs involved in different developmental stages during secondary growth were clustered using the correlation algorithm of WGCNA. Among the genes, some key TFs and genes linked to secondary growth were identified. For instance, the TFs of the MYB, NAC, and PLATZ gene families. Secondary growth in plants is regulated by a regulatory network of TFs and enzyme genes involved in secondary wall synthesis [15,23,53]. For example, the EgMYB1 interacted with the linker histone variant EgH1.3 to inhibit lignin deposition in *Eucalyptus* xylem cell walls [54]. The *PdMYB221* gene in poplar is homologous to the *Arabidopsis* R2R3-MYB TF, *AtMYB4* gene and regulates SCW biosynthesis during secondary growth [19]. In this study, genes related to MYB TFs were identified in both the xylem and phloem of *S. psammophila* stems (Appendix A). The secondary wall-associated NAC domain (SND) TFs are major wood formation regulators [23,55]. NAC and PLATZ are important regulators in the secondary growth of *Populus* stems [56]. We found several genes linked to PLATZ TFs and NAC domains by examining the genes in the secondary xylem and phloem of *S. psammophila* stems (Appendix A). In this paper, we constructed a co-expression network diagram of genes regulating NAC domain, bHLH-MYC and R2R3-MYB TFs, and PLATZ TFs. These TF genes regulate the stress tolerance, growth and development, and biosynthesis of SCW in plants, forming a regulatory network that collectively regulates the secondary growth of *S. psammophila*. The *Sapur.006G098200* gene of PLATZ TFs has the highest expression level in 2-year-old secondary xylem. The PLATZ family of TFs is important in the regulation of plant growth and adaptation to the stress environment [57]. During the secondary growth process of *S. psammophila*, the expression level of the gene may be regulated to improve the plant stress tolerance and thus regulate the plant growth.

The discovery that CSE is a central enzyme in the lignin biosynthesis pathway promotes the use of lignin [48]. *Arabidopsis thaliana cse* mutants deposited less lignin compared to the wild type, and the lignin content in *cse* mutants decreased as the levels of *p*-hydroxyphenyl units increased [48,49]. Down-regulation of the *CSE* gene led to a 25% reduction in lignin content, an increase in *p*-hydroxyphenyl units (lignin aggregates), and cellulose content in hybrid poplar (*Populus tremula* × *Populus alba*) [58]. The presence of lignin limits the effective hydrolysis of wood fibers. Wood fiber feedstocks with lower or more easily degradable lignin are more helpful in reducing the processing cost of paper [59,60]. The pulping process of papermaking usually requires chemical reagent treatment under high-temperature conditions. In addition, other substances in the plant used for pulping can also affect the inhomogeneity, hydrophobicity, and efficiency of lignin removal during the pulping process [61,62]. Wood containing less lignin can not only reduce industrial costs but also improve the quality of paper when used for papermaking. There are some genes regulating CSE in *S. psammophila*, and we can further explore the effects of these genes on the lignin content of *S. psammophila* and used the pruned branches for biomass energy production to improve the economic benefits of *S. psammophila*.

## 4. Materials and Methods

### 4.1. Plant Material, RNA Extraction, and Sequencing

The experimental material for this study was obtained from the asexual *Salix psammophila* C. Wang & C. Y. Yang ‘17-38’ from the Germplasm Resources gene bank of *S. psammophila* in Ordos Dalad, the Inner Mongolia Autonomous Region of China (110°38′59.1″ E, 40°14′15.5″ N). Select 1, 2, and 3-year-old *S. psammophila* branches with uniform growth, and take three biological replicates from the branches of *S. psammophila* at each growth stage. And quickly store the stems used to observe the cell structure of *S. psammophila* in FAA fixative (70% alcohol:formalin:acetic acid = 90:5:5) and 2.5% glutaraldehyde for at least 24 h, and the remaining parts were kept in the refrigerator [63]. Using the sliding microtome (Leica SM 2010R, Wetzlar, Germany), 16 μm thick cross-sections were cut from 1, 2, and 3-year-old *S. psammophila* stems, respectively. Stain with Astra Blue-phloroglucinol staining solution for 5 min (the lignified part turns red) and observe the anatomical structure of *S. psammophila* stems using a Nikon upright electric force microscopy (Nikon ECLIPSE Ni-E, Tokyo, Japan).

Stems (50 cm above the base) used for RNA-seq analysis were scraped with sterilized blades to obtain the secondary xylem and secondary phloem, and then snap-frozen in liquid nitrogen and stored at −80 °C. RNA was extracted from the secondary xylem and secondary phloem of *S. psammophila* using the Rapid Universal Plant RNA Extraction Kit (Cat: 0416-50gk, Beijing Xinghua Yueyang Biotechnology Co., Ltd., China) following the manufacturer’s instructions. VAHTS^®^ Universal V8 RNA-seq Library Prep Kit for Illumina (RN605, Vazyme Biotech Co., Ltd., Nanjing, China) was used to construct the 1, 2, and 3-year-old xylem and phloem RNA-seq libraries of *S. psammophila* stems. The prepared libraries were sequenced using the next-generation sequencing technology on Illumina Novaseq 6000 (Novogene, Beijing, China).

### 4.2. RNA-seq Data Analysis

Fastp (v0.24.3) [64] was used for the quality control of raw data to obtain clean reads. Clean reads were mapped to the reference genome of the *S. purpurea* genome (v5.1) from Phytozome (https://phytozome.jgi.doe.gov/pz/portal.html#!info?alias=Org_Spurpurea, accessed on 19 May 2024) using Hisat2 (v2.2.1) [65]. Then, the unique comparison reads were extracted, and Samtools (v1.6) [66] was used to convert the sam files into bam files and sorted to facilitate the calculations. Htseq-count (v2.0.5) [67] and Stringtie (v2.1.7) [68] were used to quantify the transcriptome data, and the count and FPKM values were calculated for subsequent analysis.

DESeq2 [69] was used to analyze DEGs in the xylem and phloem of 1, 2, and 3-year-old *S. psammophila* stems. Each sample was used as a control in the previous year, with the statistical criteria of *p*-value < 0.05 and ≥2-fold or ≤2-fold fold difference (FC). R package weighted gene co-expression network analysis (WGCNA) [51] was used to calculate the co-expression network.

## 5. Conclusions

To systematically elucidate the regulatory mechanism of secondary growth in *S. psammophila*, the cell structure and gene expression of *S. psammophila* at different developmental stages were analyzed. Changes in the secondary growth of *S. psammophila* mainly occurred in the 2 to 3-year-old stage, with rapid height growth and stem growth. The expression levels of lignin biosynthesis genes *CSE1* and *CSE2* showed significant changes in 3-year-old secondary xylem in *S. psammophila*. In addition, genes related to the NAC domain, bHLH-MYC and R2R3-MYB TFs, and PLATZ TFs are expressed in the secondary growth zone of *S. psammophila* and form a regulatory network. PLATZ TFs, as plant specific TFs, regulate the stress tolerance of plants. The high expression of the *Sapur.006G098200* gene of PLATZ TFs in 2-year-old secondary xylem indicates that *S. psammophila* promotes rapid secondary growth by enhancing stress tolerance. These studies have found that the related genes and transcriptional regulatory networks involved in lignin biosynthesis are of great significance for further analysis of the secondary growth regulation mechanism of *S. psammophila*.

## Figures and Tables

**Figure 1 plants-14-00459-f001:**
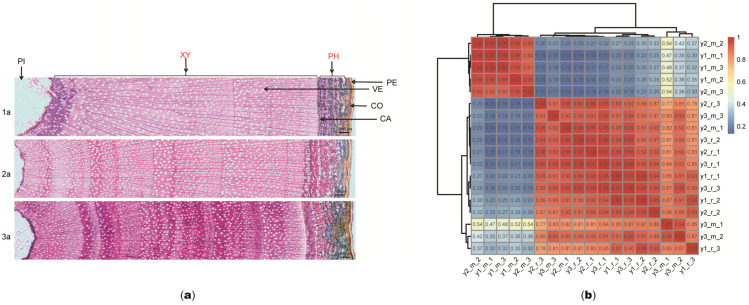
Different developmental stages of *Salix psammophila* stem. (**a**) Anatomy of stem from 1, 2, and 3-year-old *S. psammophila*. Scale bars =  500 µm. PE, peridermis; CO, cortex; PH, phloem; CA, cambium, XY, xylem; PI, pith; VE, vessel. 1a: 1-year-old S. psammophila; 2a: 2-year-old S. psammophila; 3a: 3-year-old S. psammophila. (**b**) Heatmap for intra- and intergroup sample correlation analysis of secondary xylem and phloem in *S. psammophila*.

**Figure 2 plants-14-00459-f002:**
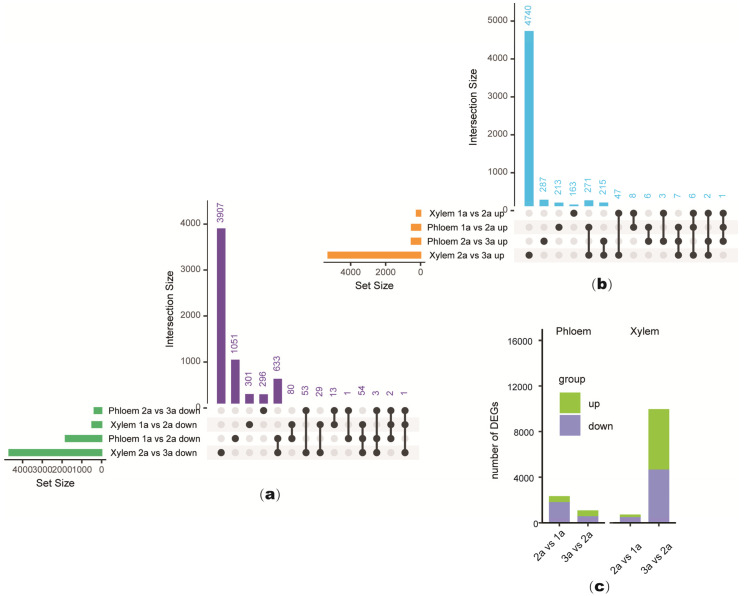
Expression of differentially expressed genes (DEGs) in secondary xylem and phloem of *Salix psammophila* stems at different developmental stages. (**a**) Venn diagram representing numbers of down-regulated DEGs and overlapping sets. (**b**) Venn diagram representing numbers of up-regulated DEGs and overlapping sets. (**c**) Histogram representing numbers of DEGs in secondary xylem and phloem of *S. psammophila* stems.

**Figure 3 plants-14-00459-f003:**
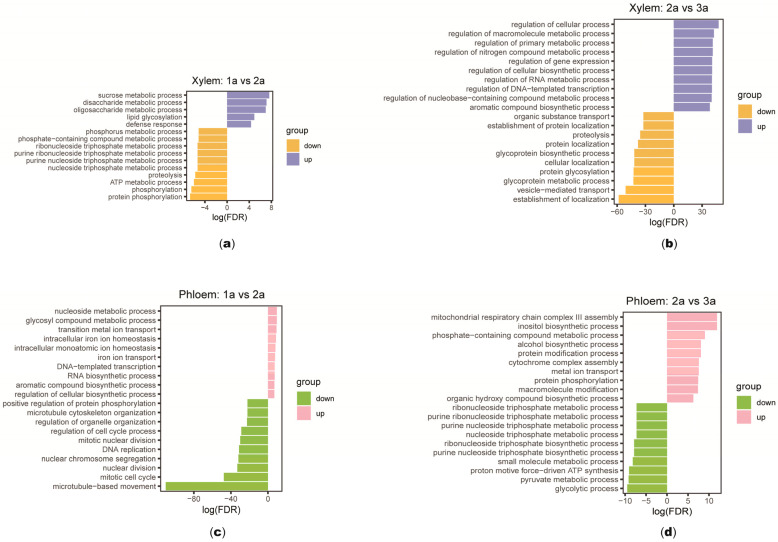
GO annotation of differentially expressed genes (DEGs) in secondary xylem and phloem of *Salix psammophila* stems. (**a**) Biological process GO annotation of DEGs in 1 and 2-year-old secondary xylem. (**b**) Biological process GO annotation of DEGs in 2 and 3-year-old secondary xylem. (**c**) Biological process GO annotation of DEGs in 1 and 2-year-old secondary phloem. (**d**) Biological process GO annotation of DEGs in 2 and 3-year-old secondary phloem.

**Figure 4 plants-14-00459-f004:**
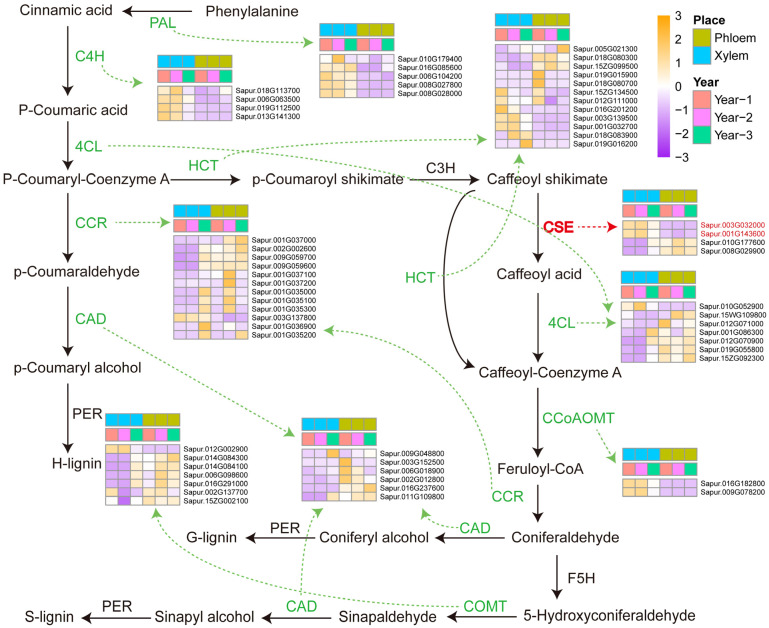
The lignin synthesis pathway map. The color scale represents the module-trait correlation from −3 (purple) to 3 (orange). The color bar indicates the expression and correlation levels from low (purple) to high (orange). The place includes the secondary xylem (blue) and phloem (tender green). The year represents the different stages of development, including 1-year-old (red), 2-year-old (purple), and 3-year-old (grass green).

**Figure 5 plants-14-00459-f005:**
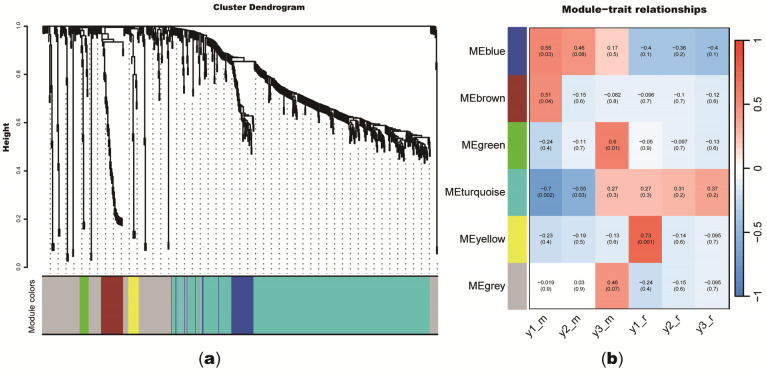
Weighted gene co-expression network analysis of genes. (**a**) Cluster dendrogram of transcription factors on expression levels in secondary xylem and secondary phloem of 1, 2, and 3-year-old *Salix psammophila* stems. Gene is represented by each branch, and gene co-expression module is represented by each color below. (**b**) Weighted gene co-expression network analysis’s module-trait association analysis. Module-trait correlations are shown by number in squares (with matching *p*-values in parentheses). Module-trait correlations are shown by color scale, which ranges from −1 (blue) to 1 (red). Expression and correlation levels are shown by color bar, which ranges from low (blue) to high (red). Panels y1_m, y2_m, and y3_m represent developmental stages of secondary xylem; y1_r, y2_r, and y3_r represent developmental stages of secondary phloem.

**Figure 6 plants-14-00459-f006:**
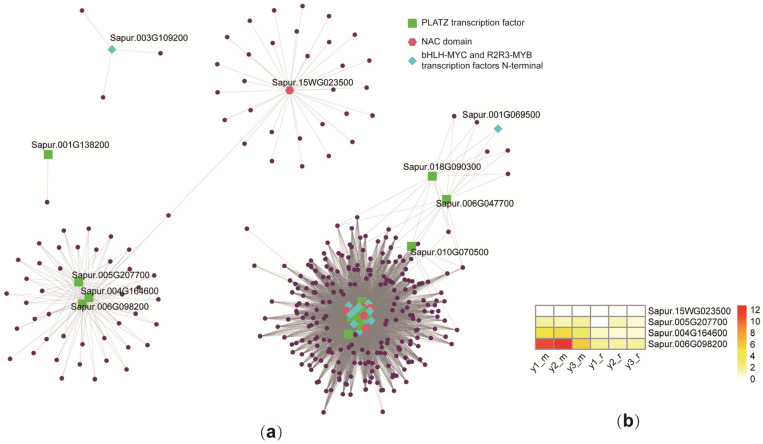
Co-expression network diagram. (**a**) Co-expression network analysis of 79 selected genes from genes in modules. Green dots represent PLATZ TFs, red dots represent NAC domain, blue dots represent bHLH-MYC and R2R3-MYB TFs, and purple dots represent NA. Linked weights between genes are shown by lines. (**b**) Genes are shown by color scale, which ranges from 0 (white) to 12 (red). Expression and correlation levels are shown by color bar, which ranges from low (white) to high (red).

**Table 1 plants-14-00459-t001:** The 1, 2, and 3-year-old *Salix psammophila* tree heights and basal stems.

	Tree Height/(m)	Average Value/(m)	Basal Stem/(cm)	Average Value/(cm)
1a_1	2.33	2.61	0.93	1.22
1a_2	2.79		1.40	
1a_3	2.70		1.33	
2a_1	3.06	2.92	1.60	1.43
2a_2	2.89		1.50	
2a_3	2.80		1.20	
3a_1	3.74	3.39	1.87	1.83
3a_2	3.10		1.83	
3a_3	3.32		1.80	

## Data Availability

The data presented in this study are available on request from the corresponding author. The raw sequence data reported in this paper were deposited in the Genome Sequence Archive (Genomics, Proteomics & Bioinformatics 2021) in the National Genomics Data Center (Nucleic Acids Res 2024), the China National Center for Bioinformation/Beijing Institute of Genomics, and the Chinese Academy of Sciences (GSA: CRA021762), which are publicly accessible at https://ngdc.cncb.ac.cn/gsa, accessed on 20 May 2024.

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
