# Peer review of "Insights into Molecular Mechanism of Secondary Xylem Rapid Growth in Salix psammophila"

_plants, 2025, doi:10.3390/plants14030459_

Round 1
Reviewer 1 Report
Comments and Suggestions for Authors
In the introduction: Please explain whether they were conducted on other species of Salix and Populus. Are these unique studies or is there already such data?
Please provide a photo of the plant being studied.
In the methodology, please provide the full name of the species being studied along with the name of the author of this species.
Please explain in detail how the histological sections were prepared and how they were stained.
What does "electric microscope" mean?
What does "vascular" mean in Figure 1?
Where is the cambium located in photos of anatomical preparations?
Reviewer 2 Report
Comments and Suggestions for Authors
The work is undoubtedly very interesting and of high quality in terms of studying the genetic regulatory mechanism of secondary growth in S. psammophila and transcriptome analysis.
The only detail I disagree with is the first section of the results entitled '', as it does not describe the characteristics of the wood in the first three years of growth. The lack of this detail detracts from the rest of the incredible work that has been done. This section should be completely rewritten, using the IAWA methodology for wood description, otherwise, use the previously published wood descriptions of the species and explain if it is not necessary to know the differences in the wood in the first three years of life.

Reviewer 3 Report
Comments and Suggestions for Authors
The manuscript is clear and relevant to the field. However, there are concerns regarding the cited references; 46% are older than five years, even when excluding publications from 2019. The results are interpreted appropriately.
The introduction should be re-written and re-structured very well to reflect the research’s objectives and significance. The hypothesis and aim should be highlighted at the end of this section.
All figures and tables are presented well, but I believe that having 18 tables and 6 figures in the supplementary material is excessive, making the reading experience more difficult. Additionally, the conclusions repeat the results. They should be phrased in a way that highlights the novelty and significance of the work.
Materials and methods. Please provide more important information about plant material (biological replications), growth, and climatic conditions.
Round 2
Reviewer 2 Report
Comments and Suggestions for Authors
The work still contains misconceptions about the secondary structure of the stem. These have been changed and corrected in the text. The main thing is that these stems no longer have epidermis, which has been replaced by peridermis.
Other minor recommendations have been added to the text. With these corrections, the paper should now be accepted.

Reviewer 3 Report
Comments and Suggestions for Authors
Thank you for considering my suggestions.